# Reducing Morbidity and Mortality Rates from COVID-19, Influenza and Pneumococcal Illness in Nursing Homes and Long-Term Care Facilities by Vaccination and Comprehensive Infection Control Interventions

**DOI:** 10.3390/geriatrics6020048

**Published:** 2021-05-08

**Authors:** Roger E. Thomas

**Affiliations:** Department of Family Medicine, Faculty of Medicine, University of Calgary, Calgary, AB T2M 1M1, Canada; rthomas@ucalgary.ca

**Keywords:** nursing homes, long-term care homes, respiratory infections, COVID-19, influenza, pneumococcal illness, comprehensive infection control interventions, individual room fresh air entry and venting, automatically triggered interventions

## Abstract

The COVID-19 pandemic identifies the problems of preventing respiratory illnesses in seniors, especially frail multimorbidity seniors in nursing homes and Long-Term Care Facilities (LCTFs). Medline and Embase were searched for nursing homes, long-term care facilities, respiratory tract infections, disease transmission, infection control, mortality, systematic reviews and meta-analyses. For seniors, there is strong evidence to vaccinate against influenza, SARS-CoV-2 and pneumococcal disease, and evidence is awaited for effectiveness against COVID-19 variants and when to revaccinate. There is strong evidence to promptly introduce comprehensive infection control interventions in LCFTs: no admissions from inpatient wards with COVID-19 patients; quarantine and monitor new admissions in single-patient rooms; screen residents, staff and visitors daily for temperature and symptoms; and staff work in only one home. Depending on the vaccination situation and the current risk situation, visiting restrictions and meals in the residents’ own rooms may be necessary, and reduce crowding with individual patient rooms. Regional LTCF administrators should closely monitor and provide staff and PPE resources. The CDC COVID-19 tool measures 33 infection control indicators. Hand washing, social distancing, PPE (gowns, gloves, masks, eye protection), enhanced cleaning of rooms and high-touch surfaces need comprehensive implementation while awaiting more studies at low risk of bias. Individual ventilation with HEPA filters for all patient and common rooms and hallways is needed.

## 1. Introduction

In healthy seniors, influenza viruses, SARS-CoV-2, respiratory syncytial virus (RSV) and human metapneumovirus (hMNV) are the most frequent causes of viral pneumonia, and adenoviruses, rhinoviruses and parainfluenza viruses are less common. Immunocompromised seniors are a highly at-risk group of patients in LTCFs and nursing homes and are at risk from the same viruses together with reactivating latent viruses, such as cytomegalovirus, herpes simplex virus and adenoviruses. A specific pathogen is not identified in a third of large surveillance studies of community-acquired pneumonia, either because peak shedding has occurred or the viral titre is low. When all nucleic acids in samples are sequenced, a clearer picture of individual and population bioviromes will be obtained. Pneumococcal pneumonia is the most important bacterial cause of pneumonia.

In this article on individuals ≥ 65 in nursing homes and LTCFs, the focus is on the five most frequent pathogens: influenza, respiratory syncytial virus, SARS-CoV-2, hMNV and pneumococcal pneumonia and invasive pneumococcal disease [1], their outcomes in seniors and frail multimorbidity seniors, and the interventions needed in nursing homes and LTCFs to minimise mortality.

Influenza is a single-stranded, segmented, negative-sense, RNA virus in the Orthomyxoviridae family. Serotypes A and C infect multiple species and B almost exclusively humans. As influenza A viruses have 18 different hemagglutinin (H) and 11 neuraminidase (N) surface protein subtypes, reassortment can cause major pandemics: H1N1 (1918 pandemic), H2N2 (1957 pandemic), H3N2 (1968 pandemic), H1N1pdm2009 (2009 pandemic) and H1N1, H3N2 and type B viruses cause annual seasonal epidemics. As up to 10% of cases of influenza pneumonia are accompanied by staphylococcal and streptococcal bacterial pneumonia, it is especially serious for the elderly with an increased risk of mortality and death within three weeks due additionally to heart failure. Pandemics can occur due to transmission from animals: influenza viruses are frequent in the gastrointestinal tracts of birds, and if they settle on a farm with pigs (whose lungs possess receptors for both human and avian influenza), the pigs can reassort the avian influenza to be infective for humans. The transmission of H5N1, H7N7 and H7N9 influenza viruses can occur from chickens to humans, with major epidemics of H5N1 in 1997 in Hong Kong and H7N9 in 2013 in China [1].

RSV is a single-stranded, negative-sense, nonsegmented, RNA virus of the Paramyxoviridae family and is a frequent cause of infection in both children and the elderly. In the elderly, RSV is an important causes of hospitalisation, with the same pattern of heart failure and requirement of major clinical support as influenza [1].

Human metapneumovirus (hMPV) is also a member of the Paramyxoviridae family and causes about 50% fewer lower respiratory tract infections than RSV, probably because it does not contain the genes that RSV expresses in infected host cells to negate the effectiveness of the patient’s immune system [1].

SARS-CoV-2 is a member of the Coronaviridae genus. The first major epidemic (November 2002 through July 2003) involved > 8000 cases with a 10% mortality rate, the Middle East respiratory syndrome coronavirus (MERS-CoV-2) epidemic in 2012 had a 35% mortality rate, and in the current pandemic more than 80% of deaths have occurred in the frail elderly. The virus is endemic in bats with the infection of civet cats in 2002-3, MERS in 2012 in bats with humans acquiring it from dromedaries, and in the current pandemic the initial exchange began with infection from bats to pangolins [1]. More than 36 animal species are known to chronically harbour coronaviruses.

The COVID-19 pandemic has provided important evidence of the problems of preventing respiratory illness in seniors, especially frail seniors with multimorbidity in nursing homes and LTCFs. In the US, the CDC reported that 80% of deaths from COVID-19 were in those >65 and there were sharply increasing rates above the age of 65 [2] (Table 1), and it has caused a pandemic comparable in worldwide spread to the 1918 influenza pandemic. The purpose of this article is to examine optimum strategies that nursing homes and LTCFs should implement to minimise mortality from all respiratory viruses.

A multi-country study compared mortality rates for different age groups from COVID-19 over a six-week period to May 8 2020 for the 21 countries with the highest recorded number of cases of COVID-19 (Austria, Belgium, Brazil, Canada, China, France, Germany, India, Iran, Israel, Italy, Netherlands, Portugal, Russia, South Korea, Spain, Sweden, Switzerland, Turkey, the United Kingdom, and the United States). Compared to COVID-19 deaths in those ≤54 years, for those 55–64 years the incident rate ratio (IRR) was 8.1 (95% confidence interval (CI) 7.7 to 8.5), for those ≥65 years the IRR was 62.1 (59.7 to 64.7), and for males the IRR was 1.77 (1.74 to 1.79) [3].

### Purposes

To identify in nursing homes and LTCFs: (1) The numbers of patients infected with respiratory tract infections, risk factors and mortality. (2) The effectiveness of vaccination against respiratory illnesses for residents in nursing homes and LTCFs and the healthcare workers (HCW) who care for them, and methods of increasing vaccination rates for SARS-CoV-2, influenza, and pneumococci which cause pneumonia and invasive pneumococcal disease. (3) The effectiveness of comprehensive interventions to reduce rates of respiratory illnesses using vaccination, hand washing, PPE (masks, eye protection, gloves, waterproof gowns), comprehensive infection control strategies such as the CDC indictors of care, a reduction in crowding in homes, the provision of isolation wings and rooms in nursing homes and LTCFs, and air flow redesign in existing and new homes to reduce the prevalence of mortality from respiratory tract infections in nursing homes and LTCFs. (4) Assess which preventive interventions function automatically and which for effectiveness are most and least dependent for the completeness of their performance on the continued diligence of the participants.

## 2. Materials and Methods

### Literature Search

Medline and Embase were searched on 4 January 2021 and again on 9 March 2021 (Table 2). All search terms were used with the .mp suffix rather than the MeSH headings as this provides a wider search. The titles and abstracts of all studies in searches 10, 12, 16, 20 and 22 were read and then relevant studies selected and read in full text.

## 3. Results

### 3.1. Frail Health Status of Nursing Home and LTCF Residents

Some seniors elect to move to lodges and senior residencies, and others need more intermittent or continued care in nursing homes. Nursing homes and LTCFs vary from low care-requirement homes where patients are provided meals with intermittent care for health problems but are expected to look after all other aspects of their care, to high care-requirement homes with most patients having high rates of frailty, comorbidity and dependence. In high care-requirement homes, nursing care intensity and difficulty are increased and involve major workloads, including coping with crises (e.g., falls, consequences of dementia, depression and aggression, new acute illnesses such as pneumonia and referring ill patients to hospital), assessing, examining and measuring physical signs, completing measuring scales for monitoring purposes and regulatory agencies, taking samples and sending them to the laboratory, data entry on computers, administering medications and treatments, discussing their care with colleagues and family members, providing food (some patients need puree food fed one spoon at a time), toileting, diaper changing, turning in bed to avoid bedsores, laundry, escorting to dining rooms, and socialising with and encouraging patients.

Respiratory pandemics such as SARS-CoV-2 or influenza markedly increase demands for skilled care and transfers to hospital. Some homes care for many very ill patients with consequent short stays to the time of their death, and a study in Norway is an example of such homes with a median survival of 2.2 years and with acuity which would require all respiratory illness reduction strategies [4]. Nursing homes in Iceland, Ireland and the US had similar median survival times between 2.3 and 2.8 years [5,6,7,8,9], a register-based study in Norway 1994–2004 2.1 years [10], and a study in France an annual mortality rate of 17.4% [11]. The acuity of care is illustrated by a systematic review of transfers to hospital for emergency care: 59% were triaged as urgent or emergent compared to 45% of all emergency department presentations and 1–5% died in the emergency department, 5–34% after admission to hospital and 12–29% within a month of discharge [12] (Table 3). Patients have high levels of chronic comorbidities [13,14].

### 3.2. Infections and Respiratory Infections in Nursing Homes and LTCFs

Systematic reviews of infections in LCTFs identified wide variations in reported respiratory illness rates, higher attack rates for viral than bacterial outbreaks, and inadequate control measures in many studies [15,16,17] (Table 4). There is considerable scope for increasing influenza vaccination rates in seniors, and for 26 million Medicare fee-for-service patents the highest rates were 57.9% in both the 80–84 and 85–89 age groups, with marked decreases in the age groups 90+ [18]. For those ≥65 in Canada, the estimated rate in 2020 was 70.3% (95% CI 66.7, 73.8) [19]. In a study in Alicante, Spain, patients hospitalised with influenza had more chronic disabilities and the mortality rate was 19% for those ≥ 65 compared to 2.9% for those <65 [20]. Vaccine effectiveness is lower as seniors increase in age and infection protection measures become increasingly important.

Studies of COVID-19 in nursing homes and LTCFs demonstrate the large percentages of patients and staff who are asymptomatic, the rapidity of the spread of epidemics from a single case [21], the necessity for proactive comprehensive control strategies instituted before any cases occur, and the imperative to institute lockdown and comprehensive control strategies immediately the first case is identified (Table 4). The rapidity of the spread of COVID-19 was illustrated in an LCTF in Washington State which identified the first case on 28 February—by 18 March there were 167 confirmed COVID-19 cases (101 residents, 50 HCWs, 16 visitors) epidemiologically linked to the facility, and hospitalisation rates for COVID-19 positive residents were 54.5%, visitors 50.0%, staff 6.0% and there were 34 deaths in residents. On 10 March 2020, the state governor implemented mandatory screening of health care workers and visitor restrictions for all LCTFs and clinical monitoring, social distancing, and restriction of resident movement and group activities [22]. Similarly, in Ireland the first case occurred on 29 February 2020, and in a national survey 18 April to 5 May 2020, in the Dublin region the confirmed COVID-19 rate for residents was 40.8% (25% asymptomatic), the case fatality rate was 25.8%, and for staff the confirmed rate was 33.6% (27.6% asymptomatic) [23].

Progression to severe disease can be rapid in older seniors with multiple comorbidities. In a retrospective cohort study of 832 consecutive COVID-19 admissions 4 March to 24 April 2020 in five hospitals in Maryland and Washington, DC, 787 patients were admitted with mild to moderate disease and 45 with severe disease (measured with the WHO scale) [24], and at discharge 523 (63%) had experienced mild to moderate disease, 171 (20%) severe disease and 131 (16%) died. Progression to severe disease or death was rapid and occurred in 181 (60%) by day 2 and 238 (79%) by day 4 [25].

Mortality rates are markedly higher for older seniors. In a national study in Norway of all individuals who tested positive for SARS-CoV-2 by the end of June 2020, for those ≥ 90 the risks of hospitalisation (RR = 9.5; 7.1, 12.7) and death (RR = 607.9; 145.5, 2540.1) were much higher than for those <50 years and the risk of death for nursing home residents was higher (RR = 4.2; 3.1, 5.7) [26]. Mortality rates vary with the quality of care and were much higher in homes in France [27] with poorer clinical care, and in California outbreak sizes were 13 times larger in for-profit than in non-profit homes [28]. Mortality rates can vary widely between regions, and the wide variations in mortality rates between Italian regions have not yet been elucidated [29].

### 3.3. Rates of Community Acquired Pneumococcal Pneumonia (CAP) and Invasive Pneumococcal Disease (IPD) in Seniors in the Community and in Nursing Homes and LCTFs

The prevalence of pneumococcal disease has been dramatically reduced in children after vaccination with PCV vaccines was implemented but remains substantial in seniors. In the US, in 2019 there were an estimated 502,600 nonbacteremic CAP cases, 29,500 IPD cases and 25,400 pneumococcal-related deaths among the 91.5 million adults > 50 years [30].

Rates of pneumococcal disease for the UK 1990–2015 were assessed by a systematic review of 38 prospective, retrospective, registry and surveillance cohorts based on patients admitted to hospital (no studies of outpatients were identified). In 2013/14, the rate of community acquired pneumonia (CAP) and non-invasive disease was 20.6/100,000 adults, and for invasive pneumococcal invasive disease (IPD) it was 6.85/100,000 for all adult age groups and 20.58/100,000 for those > 65. There were approximately 192,281 hospital admissions for pneumonia and 6000 cases of IPD in the UK in 2013/14 [31].

The incidence of pneumococcal disease in seniors is increased by several risk factors. A survey in UK general practice in 2009 estimated that in patients ≥ 65 with no risk factors, the incidence of CAP was 17.9/100,000, for the 44.8% of patients ≥ 65 who had at least one risk factor 48/100,000, for the risk factor of Chronic Obstructive Pulmonary Disease (COPD) 91/100,000, and for chronic liver disease 129/100,000. Several serotypes had high fatality rates for those ≥ 65: 3 (39%), 31 (40%), and 19F (41%) [32].

In the UK, the pneumococcal polysaccharide vaccine (PPV23) was authorised for patients ≥ 80 in 2003, ≥ 75 in 2004/5 and ≥ 65 in 2005/6. For children, the first pneumococcal conjugate vaccine PCV7 was introduced in 2007 and PCV13 in 2010. After the introduction of the PCV vaccines for children, there was a beneficial herd protection effect on rates of pneumococcal diseases in seniors. For those > 65 years, in 2008–10 the incidence of PCV13 IPD serotypes was 10.33/100,000 and declined to 3.72/100,000 in 2013/14 [33].

There is no single document which provides world-wide pneumococcal vaccination rates through 2019 for those ≥65. Pneumococcal vaccination rates in those ≥65 are suboptimal. In 2015, in the US the vaccination rate was 60.2% for those 65–74, 68.6% for 75–84 and 68.3% for 85+; 41.7% for Hispanics, 68.1% for non-Hispanic whites, 50.2 for non-Hispanic Blacks, 49% for non-Hispanic Asians; and 48.7% for poor and 66% for non-poor individuals (CDC poor and non-poor classification) [34].

A retrospective cohort of >26 million US Medicare fee-for-service patients ≥ 65 2015–2017 is one of the few studies that provides a detailed listing by quintiles of vaccination rates for those ≥65 to 100+. Pneumococcal vaccination rates were: 65–69 (47.5%), 70–74 (49.7%), 75–79 (49.7%), 80–84 (48.1%), 85–89 (45.5%), 90–94 (39.7%), 95–99 (32.5%), and 100+ (15.1%) [17]. For US nursing homes, the Minimum Data Set of the US Centers for Medicare and Medicaid Services reported pneumococcal vaccination coverage increased from 67.4% in 2006 to 78.4% in 2014, but there were large variations in pneumococcal vaccination coverage by state in 2014 (55.0% to 89.7%) [18].

For U.S. nursing home residents influenza and pneumococcal vaccination coverage increased from 2005 to 2015 but did not achieve the 90% national target for both vaccines and non-Hispanic black and Hispanic residents had lower vaccination rates [35]. A retrospective population-based observational study in January 2017 of 2,057,656 individuals ≥ 50 years old in primary care centres in Catalonia, Spain, found large variations by age in the percentages vaccinated: 796,879 (38.7%) had received PPV23, and of these 9.2% (95,409/1,039,872) of 50–64 year olds, 63.1% (434,408/688,786) of 65–79 year olds and 81.2% (267,062/328,998) of ≥80 year olds (*p* < 0.001). However, only 13,607 (0.7%) had received PCV13 [36].

In a sample of 2,531,227 individuals ≥ 15 years in the Shanghai Centers for Disease Control and Prevention information systems on chronic disease management, hospital records, and immunizations, 22.8% were vaccinated for pneumonia from January 2013 to July 2017 but only 0.4% for influenza during the 2016/17 influenza season [37].

A study of nearly 10,000 IPD cases in those 65 and older in England and Wales 2012–2016 found that PPV23 vaccination effectiveness was 27% (95% CI 17, 35) after adjusting for age, comorbidities and infection year. Vaccine effectiveness varied with the interval after vaccination, and was 41% (95% CI 23, 54) for those vaccinated within two years, 34% (15, 48) for those vaccinated 2–4 years previously, and 23% (95% CI 12, 32) for those vaccinated ≥ 5 years previously. Vaccine effectiveness was 45% (95% CI 27, 59) in those with no risk factors, 25% (95% CI 11, 37) in high-risk immunocompetent patients and 13% (95% CI 9, 30) in the immunocompromised patients (difference *p* = 0.05) [38].

## 4. Results: Improving the Health Status and Outcomes of Patients in LTCFs

### 4.1. Interventions to Increase Vaccination Rates in Seniors and HCWs

A systematic review identified 61 RCTs of interventions to increase community influenza vaccination rates. Most interventions focused on increasing demand from individuals by contacting them with letters or postcards and the most successful ones had a personal component consisting of a phone call by a receptionist or nurse. Interventions to make vaccines more available through home or group patient visits were also successful. Interventions paying physicians, competitions between physicians and benchmarking physicians’ vaccination rates to those of the top 10% of vaccinators were also successful [39] (Table 5). An RCT in 823 nursing homes in the US used high-dose vaccinations to compensate for the waning of immunocompetence in seniors and found that the relative risk of influenza was reduced with high-dose vaccinations (adjusted relative risk (RR) = 0.873; 95% CI 0.776, 0.982; *p* = 0.023) [40].

National vaccination programmes are an important stimulus to increase community pneumococcal vaccination rates [41,42], but rates still remain below national targets for seniors. Even after campaigns with subsidised vaccination in Japan, the national rate was only 74%. In England, the coverage of PPV23 in those ≥65 was similar to Japan at 70.1% in 2015 and 69.5% in 2018 [43]. In Australia, after public funding for PPV23 commenced in 2005, the vaccination rate increased from 35.4% before 2005 to 56.0% after 2005 [44]. In South Korea, during a 20-month national immunisation program the pneumococcal vaccine rate for ≥65 years increased from 5.0% to 57.3% [45].

The WHO is supporting interventions to vaccinate all citizens in all countries against SARS-CoV-2 with the multibillion COVAX vaccine sharing programme, with the additional purpose of limiting the emergence of more variants. Seniors and front-line health workers have been prioritised in all WHO regions for SARS-CoV-2 vaccination. The duration of antibodies is uncertain and seniors will need to be monitored for antibody levels and the interval to optimum revaccination assessed. SARS-CoV-2 may require annual revaccination like influenza and shorter revaccination intervals may emerge especially with the emergence of more variants. SARS-CoV-2 and increasingly its variants will likely become an integral component of the respiratory infections that present major risks for mortality.

Finland is the only country which has made influenza vaccination compulsory for HCWs, on the analogy that multiple vaccinations are required (e.g., measles, mumps, hepatitis A and B…) to work in hospitals and clinics. The hesitancy of HCWs in other countries is based on the paucity of evidence and the reluctance of HCWs to be vaccinated (the reasons most frequently provided are lack of confidence in the effectiveness of the vaccine, concern for side-effects, dislike of needles and belief they are not at risk).

A Cochrane collaboration systematic review of vaccinating HCWs against influenza who care for seniors in nursing homes and LTCFs [46,47] identified only four c-RCTs [48,49,50,51] and did not show convincing evidence of benefit for patients for laboratory-proven influenza (low quality evidence), lower respiratory tract infections (moderate quality evidence), admissions to hospital (low quality evidence), or deaths from lower respiratory tract illness or from all causes (very low-quality evidence). The Hayward c-RCT [50] was excluded from the computations in later editions of the Cochrane review [47] because it used influenza-like illness (ILI) as the outcome measure and a systematic review of all studies showed that less than 25% of all patients identified by physicians as ILI had a positive laboratory test for influenza [52]. No c-RCTs have been published subsequently. The authors of the Cochrane review concluded that large well-conducted c-RCTs were needed with arms which also measured the effects of face masks, handwashing and high vaccination rates of HCWs [47]. The effects of social distancing should now also be added. The thoroughness of the implementation of each of these preventive measures would be essential to the success of the c-RCTs in providing evidence of high quality [53]. A systematic review identified 46 studies of interventions to increase influenza vaccination rates in HCWs but conducted a meta-analysis incorrectly pooling the nine c-RCTs, two RCTs and 35 before-and-after studies together. Of the c-RCTs, eight had unclear randomisation, five no baseline vaccination rates and six incomplete data, and the review does not provide pooled data at low risk of bias [54].

Subsequent authors have agreed with the assessment of the Cochrane reviews. The article by De Serres [55] also focused on the inappropriateness of ILI and all-cause mortality as outcome measures and the implausible greater reductions in influenza for less influenza-specific outcomes:

“In attributing patient benefit to increased HCW influenza vaccine coverage, each cRCT was found to violate the basic mathematical principle of dilution by reporting greater percentage reductions with less influenza-specific patient outcomes (i.e., all-cause mortality > ILI > laboratory-confirmed influenza) and/or patient mortality reductions exceeding even favourably derived predicted values by at least 6- to 15-fold.” [55], p. 2. Nevertheless, the authors also concluded: “Although current scientific data are inadequate to support the ethical implementation of enforced HCW influenza vaccination, they do not refute approaches to support voluntary vaccination or other more broadly protective practices, such as staying home or masking when acutely ill.” [55], p. 2.

Similarly, an “Expert Commentary” article assessing the WHO guidelines for the prevention of influenza in LTCs [56] commented:

“Although the currently available evidence may be weak for HCW vaccination to protect the frail and elderly, there is also generally no evidence against it. Therefore, it remains a biologically plausible intervention to provide individual protection to the HCW, act as a barrier against spread of infection and to help reduce the risk associated with influenza infection and prevent staff absenteeism. However, poor vaccine uptake by HCWs has been well documented. In Europe, coverage of HCWs (including those working in LTCFs) varies between countries and is generally much lower than for other vaccination targeted groups, ranging from 9.5% to 75% with a median vaccination coverage rate of 28.6%. In the United States, vaccination rates of 50–70% have been reported for LTCF workers, with coverage consistently lower than among staff working in hospital settings. Reasons given for declining vaccination include fear of side effects, lack of concern or perception of risk, doubts about vaccine efficacy, lack of availability or inconvenient delivery of vaccine, avoidance of medications and dislike of injections. Although mandatory vaccination is effective if it can be implemented, it is not legally enforceable in all countries and settings, and infection rates after the implementation of mandatory vaccination have not been studied.” [57].

Behaviour theories have been used to predict HCW influenza vaccination rates and a review of ten studies assessed them at moderate risk of bias due to self-report of vaccination and non-representative samples. Five studies used either the Health Belief Model, the Theory of Planned Behavior, the Risk Perception Attitude, or the Triandis Model of Interpersonal Behavior, which measured attitudes about the efficacy and safety of influenza vaccination, risk and benefit perceptions for the HCW and others, cues for action, and social-professional norms. These factors and sociodemographic variables predicted 85–95% of HCW influenza vaccination uptake. Vaccination in previous years was an important predictor. RCTs comparing the outcomes for patients of interventions based on these theories, mandatory vaccination policies and HCW remuneration for vaccination are needed [58].

### 4.2. Implementation of Comprehensive Infection Control Policies for COVID-19 in Nursing Homes and LTCFs

The effectiveness of comprehensive infection control strategies for respiratory infections in nursing homes and LTCFs has been best illustrated for COVID-19. The CDC has published specific recommendations for assessing the signs and symptoms of SARS-CoV-2 and influenza in nursing homes and LTCFs [59] (Table 6).

The CDC provides a guide to assessing COVID-19 and influenza signs and symptoms which can be used as an initial guide to decide risks and thus testing and isolation management, but the CDC noted the considerable similarities in symptoms [60]. The WHO has published public health advice which is more general in scope about controlling SARS-CoV-2 infections in the community, which is the reservoir for infections in nursing homes and LTCFs [61].

Two key problems in managing SARS-CoV-2 infections are the high rate of asymptomatic or minimally symptomatic patients and which management decisions to make if initial laboratory tests are negative. Prompt diagnosis of whether patients have SARS-CoV-2 permits isolation or discharge to free up clinical resources for other patients. In hospitals in Boston, Massachusetts, 2443/3358 (73%) inpatients were assessed as possibly SARS-CoV-2 but had a negative nucleic acid amplification test (NAAT). The CORAL tool (COvid Risk cALculator) uses structured entry of clinical data, and after its use there were significant reductions in patients detained for repeat SARS-CoV-2 nucleic acid amplification tests (NAATs) (54% vs. 67%; aOR 0.53, 95% CI: 0.44–0.63, *p* < 0.01), the duration of potentially infected status (adjusted difference: −19.5 (SE 1.9) hours/patient; *p* < 0.01) and average infectious disease physicians’ work-hours (adjusted difference: −57.4 (SE 2.0) hours/day; *p* < 0.01). After CORAL advised the discontinuation of precautions, no patient had a positive NAAT within the next seven 7 days [62].

The importance of comprehensive testing to identify asymptomatic cases has been demonstrated in several studies (Table 7). In Catalonia, Spain, daily monitoring for COVID-19 with a COVIDApp of 10,347 patients and ~4000 HCWs in 196 care centers identified a large number, 5090/10,347 (49%), of asymptomatic patients. There were 854 (8.3%) deaths (of which 44.8% were either suspected or confirmed COVID-19 cases), the number of high-risk long-term care facilities decreased from 19/196 (9.5%) to 3/196 (1.5%) but the number of HCWs with suspected COVID-19 remained ~ 1000 and the number isolated at home varied between 400 and 600 [63]. In 11 LTCFs in Maryland, USA, after the index case the public health department identified 153 cases within the next 20 days. In a subsequent study, all 893 untested residents were tested and an additional 354 (39.6%) tested positive for SARS-CoV-2, of whom 281 (55.4) were asymptomatic (symptoms were defined as any fever >99 °F, cough, diarrhea, respiratory decompensation, or other acute clinical status changes) [64].

Rapid and comprehensive responses to SARS-CoV-2 have been reported by several LCTFs. The key interventions are daily comprehensive monitoring of all residents, staff and visitors for symptoms and contacts using the CDC tools [65,66], and if risks are detected prompt testing using nucleic acid detection or SARS-CoV-2 antigen detection assay. As the antigen test has a lower sensitivity, confirmation of the antigen test with the SARS-CoV-2 nucleic acid detection assay is required. Due to the high rate of asymptomatic individuals, it is important to prevent admissions from institutions or communities with SARS-CoV-2 cases. Comprehensive training of staff in hand sanitation, surface sanitation, PPE (masks, eye protection, gloves, waterproof gowns) and social distancing and furloughing infected or potentially infected staff are crucial. It is important that a team of infectious disease specialists and infection preventionists closely monitors the situation daily and ensures preventive measures are followed and resources are provided promptly. Examples are the strategies of the US Veterans Affairs Midwest network, which used seven strategies to avoid the admission of SARS-CoV-2 patients and reported no positive RT-PCR SARS-CoV-2 tests from 6 March through 1 September 2020 [67]. In British Columbia, Canada, 75 LTCFs used eight strategies to minimise SARS-CoV-2 admissions and transmission within homes and there was a 32% reduction in cases [68]. In Georgia, USA, 24 LTCFs used the comprehensive set of CDC COVID-19 indicators to identify why COVID-19 rates differed between high (62% infection rate) and low prevalence (15% infection rate) homes [69]. In 17 nursing homes in France, 794 staff voluntarily confined themselves to their facilities with their 1250 residents, and COVID-19 was identified in only 5 residents (0.4%) compared to 30,569 (4.44%) in a national survey of 9513 homes [70] (Table 8).

Reducing the effect of crowding on increasing rates by providing patients with single rooms is important. For 618 nursing homes in Ontario, Canada, in homes with a low crowding index COVID-19 incidence was 4.5% and in high crowding homes 9.6%, and COVID-19 mortality was 1.3% and 2.7% [71]. Crowding in the community also increases the risk of admitting infected patients. The effects of community crowding were demonstrated in a study of the 175 postal zip codes in New York. Average household size independently explained 62% of the variation in COVID-19 rates. Rates were higher in those zip codes with higher percentages of individuals ≥ 65 or living below the poverty line, African Americans and Hispanics. Housing density in itself was not a predictor of COVID-19 rates [72].

Preventing staff working in multiple homes reduces COVID-19 rates. In 623 LTCFs in Ontario, Canada, before a policy was introduced to prevent multiple site working, 266 (42.7%) homes had staff working in at least one other home, and after the policy was instituted 79 (12.7%) had staff working in at least one other home, and the average number of connections between homes declined from 3.90 to 0.77 (*p* < 0.001) [73].

It is also important to identify which patients with SARS-CoV-2 are at high risk of poor outcomes and death.

A study in 351 US nursing homes of 5256 residents with severe acute respiratory syndrome SARS-CoV-2 found that compared to those 75 to 79 years, the all-cause mortality rate within 30 days increased with higher ages: for those 80 to 84 years OR = 1.46 (1.14 to 1.86), for those 85 to 89 years OR = 1.59 (1.25 to 2.03) and for those ≥90 OR = 2.14 (1.70 to 2.69). Four physical findings were related to a higher mortality risk: fever OR = 1.66 (1.41 to 1.96), shortness of breath OR = 2.52 (2.00 to 3.16), tachycardia OR = 1.31 (1.04 to 1.64), and hypoxia OR = 2.05 (1.68 to 2.50). Four co-morbidities were associated with an increased risk of mortality: diabetes OR = 1.21 (1.05 to 1.40), chronic kidney disease OR = 1.33 (1.11 to 1.61), severe cognitive impairment OR = 2.79 (2.14 to 3.66), and severe impairment in physical function OR = 1.64 (1.30 to 2.08) [74].

### 4.3. Increasing the Use of Medical or Surgical Masks and Hand Hygiene to Reduce the Transmission of Respiratory Viruses

The most comprehensive and recent systematic review of mask-wearing and hand hygiene is the Cochrane review by Jefferson et al. (2020), but it included no studies during the COVID-19 pandemic period [75]. In many studies, the risk of bias for the RCTs and c-RCTs was mostly high or unclear. The review included only three c-RCTs of hand hygiene in nursing homes. McConeghy’s 2017 study [76] was assessed at unclear risk for random sequence generation and allocation concealment, high risk for blinding of personnel and participants, outcome assessment and selective reporting; Temime’s 2018 study [77] was at high risk for random sequence generation, unclear risk from allocation concealment, high risk from blinding of participants and personnel, performance assessment and incomplete data; and Yeung’s 2011 study [78] was at unclear risk from random sequence generation, allocation concealment and selective reporting and high risk from blinding of participants and personnel and outcome assessment. There is only one c-RCT of hand hygiene for patients and staff which assessed bacterial contamination and there was a significant decrease in bacterial colonisation [79]. A 2020 systematic review of hand washing, distancing and mask wearing to prevent the transmission of SARS-CoV-2 identified 172 observational studies in 16 countries but no RCTs, and a meta-analysis of 44 non-randomised studies identified substantial protection by distancing more than one metre, hand washing, mask wearing and eye protection [80] (Table 9).

A study in an emergency department demonstrated a reduction in cleaning needs after contamination by potential SARS-CoV-2 patients [81]. Four modelling studies demonstrated how to minimise the contamination of airflows in respiratory disease isolation rooms. One found that directing fresh airflows from the ceiling over the patient’s bed and exhausting airflows through the wall behind the bed caused the least contamination of a manikin representing an HCW standing in front of the patient to provide care [82]. A study of contamination from door opening as a manikin representing an HCW entered from an anteroom found sliding doors reduced contamination compared to opening doors [83]. A study of pharmaceutical cleanrooms found that with the door from the anteroom closed, no particles entered the cleanroom due to the overpressure of 15 Pa and few particles entered with door opening and closing. When an individual entered the cleanroom walking fast at 1 m/s with low air flows (210 L/s), contamination was tripled compared to a lower speed of 0.5 m/s and a high air flow of 580 L/second [84]. Another study using a particle generator showed a marked increase in particle concentration within a patient room with the aerosol simulating coughing. The highest particle migration rate from the patient room to the anteroom was for particles < 3 μm (simulating SARS-CoV-2) compared to > 3 μm. A plastic barrier in the anteroom even without HEPA filters prevented the spread of 80% of particles and HEPA filters markedly reduced particle counts in the anteroom and hallway [85].

Several studies have demonstrated how large rooms or a group of rooms in a medical facility can rapidly be converted into isolation facilities by changing air flows [86,87,88,89,90,91]. In a skilled nursing facility in Lancaster, Pennsylvania, USA, an existing HVAC system was modified to minimise disease transmission between residents and staff and maintained an average pressure differential of −2.3 Pa (SD = 0.12 Pa) with the external hallway, and no transmission of SARS-CoV-2 between residents isolated to the space or transmission to the staff or other residents occurred [92].

There is very limited information on the contamination of surfaces and food in LTCFs [93,94,95]. A systematic review estimated that COVID-19 is not detectable on steel or plastic after seven days, glass after four days and wood after one day [96]. Another systematic review of 26 studies up to 4 February 2021 of the contamination of surfaces by SARS-CoV-2 RNA in hospitals analysed 3101 samples acquired before disinfection. For the nine studies with 100 or more samples, the rate ranged from 1.4% to 19%. No summary was provided for all 26 studies, but summarising the authors’ Table 1, there were nine studies with RNA samples on 0–5% of the tested surfaces, six studies with 6–10%, seven studies with 11–20%, five studies with 21–40%, three studies with 41–60% and four studies with 61–100% [97]. A c-RCT used UVC lamps in rooms previously occupied by patients with methicillin-resistant *staphylococcus aureus*, vancomycin-resistant enterococcus, *C. difficile*, or multidrug-resistant *Acinetobacter*. Rooms cleaned with quaternary ammonium cleaners (QUAD) had 51.3 cases of the targeted organisms/10,000 patient days, and for UV added to QUAD the incidence in exposed patients was 33.9 cases/10,000 exposure days, (relative risk (RR) 0.70, 95% CI 0.50–0.98; *p* = 0·036) [98,99]. A non-randomised study found a 90% reduction in viral counts within six minutes and 99.5% in 11 minutes [100]. The effects of UV-C need more modeling particularly with attention to the effectiveness of decolonisation with distance from objects [101].There is evidence that copper surfaces reduce SARS-CoV-2 contamination levels [102,103,104,105,106,107].

An important issue is identifying risks to LTCFs from being situated in communities with higher SARS-CoV-2 rates to which potential new residents, visitors and HCWs are exposed. A study used the US Centers for Medicare and Medicaid Services database of 14,886 nursing homes to identify nursing homes at risk of higher rates of COVID-19 and thus needing prioritising to promptly implement comprehensive COVID-19 prevention interventions. The nursing homes at risk had more residents; had received more Medicaid penalties, health deficiencies and deficiencies related to infections; had lower total staff and aide numbers; and were located in counties with higher COVID-19 rates, higher density communities, more residents in nursing homes, minorities, unemployed individuals and persons living below the poverty line (all *p* < 0.001). Health departments need to prioritise such homes for interventions to minimise respiratory disease outbreaks and disease rates [108].

In the Netherlands, sewage outflows from communities were sampled and COVID-19 was demonstrated in some communities days before COVID-19 was detected in any individual. The Dutch National Institute for Public Health and the Environment (RIVM) samples 318 waste-water treatment plants weekly for coronaviruses. However, there are wide variations in counts between plants depending on how many individuals use a sewage system, and dilution by rain and by industrial water. If PCR tests on humans are performed daily and on sewage weekly, the sewage tests may provide confirmation rather than early detection. Currently, the sewage tests are performed daily and are sent to a central laboratory [109].

## 5. Discussion

Worldwide air travel and tourism will increase the extent and rapidity of the spread of respiratory infections and their variants. Viruses and bacteria are widely spread in multiple non-human hosts, and many birds are infected with viruses and distribute them in annual migrations involving many thousands of miles. Many birds and animals are kept in very close proximity in live markets in some countries. More pandemics and more frequent global pandemics are likely.

Seniors and especially seniors with multi-morbidities have incurred more than 80% of the mortality in the current SARS-CoV-2 pandemic. Sars variants are emerging in Brazil, South Africa and the UK, are more infectious, also affect younger populations more, and constitute a larger proportion of all Sars cases. Many countries are highly motivated with the COVAX strategy to vaccinate their entire populations, but there are groups who do not plan to be vaccinated and a major effort to dialogue with them and understand their reasons is essential. Vaccination rates for pneumococcal disease and influenza are suboptimal in most countries. Large RCTs are needed with health workers to assess the effects on the patients they care for of optimal hand washing, social distancing, vaccination and PPE in countries where they are provided.

Comprehensive preventive interventions need to be implemented in nursing homes and LTCFs as described in the text, with rigorous monitoring with the CDC COVID tools for the comprehensive implementation of preventive interventions. Patients need to be provided with single rooms with their own air system, and hospitals structured to vent potentially infected air from each patient room, hallway and common room externally with microbiological sampling of the air and high touch surfaces. Automatic monitoring for the current vaccination status of residents, HCWs and visitors should be implemented by linking to national vaccination registers. Regular testing of patients and HCWs, high rates of vaccination, handwashing and PPE (gloving, mask wearing, eye protection and gown wearing, if these are provided) should be encouraged because current studies demonstrate that these interventions are often inadequately used and increased vaccination results in lower transmission rates.

## 6. Conclusions

Individuals ≥ 65 have high rates of morbidity and mortality due to COVID-19 infections and those in nursing homes and LTCFs are frail, have multiple co-morbidities and are especially vulnerable. Interventions are needed to reduce these rates.

### 6.1. Detection

Rapid detection of the first case of COVID-19 or other respiratory pathogens in a community and in nursing homes and LTCFs is very important in instituting a prompt comprehensive infection control plan with daily monitoring of staff and residents for symptoms, testing of patients and staff on a regular schedule, and contact tracing. As at the beginning of an outbreak there may be no symptomatic individuals in a community, an environmental approach to detect early clues of COVID-19 in a community may give useful warning by testing sewage for viruses.

### 6.2. Identification of Nursing Homes Most in Need of Help Implementing Comprehensive Infection Control Plans

Nursing homes vary greatly in the level of care they provide and this is likely to be reflected in their infection and mortality rates. The COVID-19 CDC tool is a comprehensive list of quality of care indicators which identifies nursing homes in a region most in need of help to control infections [61].

### 6.3. Implementation of Comprehensive Infection Control Plans

Comprehensive plans involve closing the home to new admissions from potentially infected individuals, contact and droplet precautions for all confirmed, suspected or exposed COVID-19 cases, quarantining patients with symptoms in isolation wards until they can be tested as positive or cleared, universal personal protective equipment (PPE) for all staff, visitors, all residents and individuals entering any unit screened daily for temperature and symptoms, admitting visitors who have been vaccinated and according to current local regulations, and cohorting staff to provide care only on specific units. The plans need to be implemented by dedicated teams of infection control specialists for the entire group of nursing homes in a community and the rapidity and completeness of implementation need to be closely monitored.

### 6.4. Restructuring Nursing Homes to Reduce Crowding

Higher crowding indices (four patients/room compared to single rooms) correlate with higher infection rates. The structural conditions should be adapted in order to be able to react more quickly to new outbreaks and new pandemic situations, e.g., due to new variants of SARS-CoV-2 or other rapidly transmissible pathogens. Ideally, each patient should have their own bathroom and toilet and an anteroom at the entrance to their room for washing and personal protective equipment so that staff and visitors can prevent the transmission of infection.

### 6.5. Upgrading Ventilation Systems in Nursing Homes

All patient rooms, hallways and common rooms need to have their own fresh air supply from an external source without recirculation and exhaust to the outside, with fresh air entering over the patient’s bed and exhausting behind the bed to avoid flowing over HCWs standing in front of them providing care. Infected or potentially infected patients need to be transported on isolation stretchers with their own air supply.

### 6.6. Increasing Influenza, Pneumococcal and SARS-CoV-2 Vaccination Rates in Seniors

Vaccination rates for influenza and pneumococcal disease are inadequate, especially in older seniors. Worldwide, there are vaccination plans for the comprehensive vaccination of seniors against coronaviruses, but there are individuals who are hesitant to be vaccinated. There are limited data on the effectiveness of these vaccinations in older people and additional efforts to protect them from potential infection are needed.

### 6.7. Identifying Interventions Programmed to Function Automatically

The intervention least dependent on the training, motivation and close supervision of patients and staff is to upgrade the ventilation systems in nursing homes so that each patient has an individual room and that all patient rooms, hallways and common rooms receive fresh air from the outside and vent the air to the outside. Due to ethical issues in exposing volunteer subjects to infections, research has involved models of hospital rooms with manikins and visualised air flows. A c-RCT implementing individual ventilation with external air entry and venting for each patient room, common room and hallway in LTCFs and microbiological sampling is needed.

Automatic monitoring of the current vaccination status of patients, staff and visitors by linking to national vaccination registries would be very helpful. Training and monitoring of standard infection reduction strategies with personal protective equipment (handwashing, mask wearing, gloving and eye protection) and especially of the WHO’s five steps in the prevention of transmission of infections would be most useful. Daily monitoring of infectious disease infection rates in communities and the LTCFs within them to trigger comprehensive infection control policies is also an important intervention. Clear public health and political responsibility for closely monitoring and assisting implementation is necessary.

## Figures and Tables

**Table 1 geriatrics-06-00048-t001:** COVID-19 hospitalisation and death rates, USA, for age groups 18–29 to 85+.

Age Group	Hospitalisation Rate	Death Rate
18–29 years	Comparison Group	Comparison Group
30–39 years	2x higher	4x higher
40–49 years	3x higher	10x higher
50–64 years	4x higher	30x higher
65–74 years	5x higher	90x higher
75–84 years	8x higher	220x higher
85+ years	13x higher	630x higher

**Table 2 geriatrics-06-00048-t002:** Literature searches 9 March 2021 in Medline and Embase.

	Search Term	Medline	Embase
1	coronavirus.mp.	81468	125824
2	Sars-CoV-2.mp. or exp SARS-CoV-2/	68254	35064
3	Covid-19.mp.	107186	96515
4	1 or 2 or 3	125192	131716
5	nursing home.mp.	22862	63414
6	homes for the aged.mp.	14435	746
7	long term care.mp.	39334	142648
8	long term care facilities.mp.	4389	5579
9	5 or 6 or 7 or 8	67177	195407
10	4 and 9	722	1288
11	mortality.mp.	1206828	1564538
12	10 and 11	186	383
13	Disease transmission	40060	107336
14	disease transmission, infectious.mp.	10443	58
15	respiratory tract infections.mp.	47170	22062
16	negative pressure isolation.mp.	73	94
17	systematic review.mp.	206869	364785
18	meta-analysis.mp.	208052	312070
19	17 or 18	318737	513093
20	10 and 19	9	27
21	13 or 14 or 15	87017	129231
22	10 and 21	37	42

**Table 3 geriatrics-06-00048-t003:** Numbers of patients in nursing homes and LCTFs, disabilities, mortality rates and respiratory infections.

Numbers of Patients in LCTFs and Nursing Homes, Disabilities and Mortality Rates		
Author, Date	Setting	Disabilities and Mortality Rates
Vossius 2018 [4]	**47 small and large nursing homes in urban and rural areas in 4 Norwegian counties** followed for 3 years. Average age 84.5 years, 83.9% dementia at baseline. **Assessment:** Trained healthcare workers (74% registered nurses, 2 days of training) collected data using structured interviews with patient and caregiver, supervised by 10 research nurses (5 days training). Dementia assessments by 2 psychiatrists (adjudicated by a 3rd), ICD-10, Clinical Dementia Rating Scale (CDR), neuropsychiatric symptoms by the Neuropsychiatric Inventory nursing home version (NPI-NH), self-care by the Physical Self-Maintenance Scale (PSMS), general health by General Medical Health Rating (GMHR), comorbidities by Charlson’s comorbidity index.	**Survival:** median Kaplan–Meier survival 2.2 years (95% CI 1.9 ± 2.4) with stable median yearly mortality 31.8%. Assessed mortality rate may be an underestimate. **Hazard ratios for mortality:** Charlson comorbidity index (HR 1.13; 1.06, 1.22; *p* <0.001), physical self-maintenance (PSMS scores) (HR 1.07; 1.03, 1.12; *p* = 0.001), age (HR 1.04; 1.01, 1.06; *p* = 0.002), residing on a nonspecialised ward (with more patients) (HR 1.03; 1.01, 1.05; *p* = 0.016).
Dwyer 2014 [12]	**Systematic review of 83 studies of emergency transfers to hospital** of residents of LTCFs ≥ 65 years.	**Triage assessment in emergency departments:** 59% triaged urgent or emergent compared to 45% of all emergency department presentations. **Reasons for admission:** multiple illnesses: respiratory tract infections (12–37% of all presentations), other infections (6–24%), falls (12–23%), fractures and orthopedic injuries (7–24%), cardiovascular illness (11–28%), altered mental state (7–12%). **Mortality:** 1–5% died in the emergency department, 5–34% after admission to hospital and 12–29% within a month of discharge.
Canadian Institute of Health Information 2020 [13]	**Numbers of nursing home patients** in Canada: 2019–2020 there were 189,662 residents in 1318 nursing homes; average age 83 years, 54% ≥ 85, 65% female. **Risk factors:** Cognitive Performance Scale (CPS) ≥4, Index of Social Engagement (ISE) ≤ 2, the Aggressive Behaviour Scale (ABS) ≥1, and the Pain Scale ≥ 2.	**Morbidities:** 61.6% dementia, 59% hypertension, 24.5% signs of depression, 24.8% diabetes, 9% cancer, 77% some urinary incontinence, 59% some bowel incontinence, 42.9% little or no social engagement, 12% total dependence for Activities of Daily Living (ADLs), 9% daily pain.
Harris-Kojetin 2018 [14]	**Numbers of nursing home patients:** 1,347,600 residents in 15,600 nursing homes. **Multiple types of long-term care:** In US in 2016 there were 65,600 remunerated regulated long-term care services, providing care for >8.3 million people in five sectors: estimated 286,300 individuals in 4600 adult day services centers, 811,500 residents in residential care communities, 1,426,000 patients receiving services from 4300 hospices and residents in 28,900 assisted living residential care communities. In 2015, ~4,455,700 patients discharged annually from home health agencies.	**Morbidities in nursing home patients:** 72% hypertension, 48% dementia, 46% depression, 38% heart disease, 32% diabetes, 26% arthritis, and 12% osteoporosis. **Hours of staff care time:** If staff used every hour for patient care, in nursing homes daily per patient RNs could provide 0.54 hours of care, licensed practical or vocational nurses 0.85, aides 2.4, social workers 0.08; in residential care, 0.2, 0.17, 2.27 and 0.03 h, respectively.

**Table 4 geriatrics-06-00048-t004:** Respiratory infections in nursing homes and LTCFs.

Infections and Respiratory Infections in LTCFs and Nursing Homes		
Author, Date	Setting	Disabilities and Mortality Rates
Lee 2020 [15]	**Systematic review of 37 studies of infections in LCTFs:** Risk of bias assessed with Risk of Bias Assessment tool for Nonrandomized Study (RoBANS) [16]. Only 6 studies at low risk for all criteria, with problems with recall bias and self-reported measurement in 7 studies, problems with confounders in 4 studies and missing data in 5 studies. No meta-analysis performed.	**1332 infection outbreaks:** most commonly reported pathogens influenza and Group A streptococcus. In 29 studies median attack rate 15.7% (8.3% for bacterial and 19.3% for viral outbreaks); 25 studies identified causes, half documented person-to-person transmission (involving poor hand hygiene and decontamination), only 9 promptly involved public health authorities, 5 studies reported creation of outbreak control teams; 60% of studies reported cases among staff, few studies implemented work restrictions.
Childs 2019 [17]	**Systematic review of 26 articles reporting respiratory infections in LCTFs:** in unvaccinated residents ≥ 60 years in LCTFs 1964–2019 to assess burden of respiratory infections in unvaccinated residents; average ages 70.8 to 90.1 years.	**Respiratory infection incidence and prevalence rates in LTCFs:** varied widely and attributed partly to seasonality. Influenza incidence rates ranged from 5.9% to 85.2%, RSV incidence 1.1% to 13.5%, pneumonia incidence rates 4.8% to 41.2%. Policy recommendations need to be based on well-designed epidemiologic studies in large populations with assessments for seasonality and risk factors in specific homes and populations.
**Influenza Rates in the Community and in LCTFs**		
Shen 2019 [18]	**Retrospective cohort study of >26 million US Medicare fee-for-service patients ≥ 65:** 2015–2017	**Influenza vaccination by age quintiles:** 65–69 (44.2%), 70–74 (52.2%), 75–79 (56.3%), 80–84 (57.9%), 85–89 (57.9%), 90–94 (54.8%), 95–99 (49.9%), and 100+ (35.8%) [20]. For US nursing homes the Minimum Data Set of the US Centers for Medicare and Medicaid Services found influenza vaccination coverage increased from 71.4% in the 2005–2006 influenza season to 75.7% in the 2014–2015 season, but there were large variations by state in influenza vaccination coverage (50.0% to 89.7%) in the 2014–2015 influenza season.
Public Health Agency of Canada 2021 [19]	**National survey of influenza vaccination rates, Canada:** 2019–2020	**Vaccination rate by age groups:** for 18–64 34.1% (95% CI 31.8, 36.5); 18–64 with serious health conditions 43.6% (95% CI 39.0, 48.1); ≥65 70.3% (95% CI 66.7, 73.8).
Ramos 2016 [20]	**Retrospective study of 219 influenza patients:** admitted to General University Hospital of Alicante, Spain, 1 January to 31 April 2015, diagnosed with influenza by molecular biology tests.	**Risk factors for patients ≥ 80 compared to those < 65:** had lower average glomerular filtration rates (49.7 mL/min vs. 62.2 mL/min; *p* = 0.006), higher rates of non-invasive mechanical ventilation (22% vs. 9.3%; *p* = 0.02), higher rates of cardiac insufficiency (40.5% vs. 16.4%; *p* < 0.001), chronic renal disease (32.9 vs. 20%; *p* = 0.03), and mortality (19% vs. 2.9%; *p* < 0.001; adjusted OR 9.2 (95% Confidence Interval [CI] 1.65 to 51.1)).
**Coronavirus in LCTFs**		
Shi 2020 [21]	**Retrospective COVID-19 cohort study in LCTF:** March 2020 of patients in an academic long-term chronic care facility in Boston, USA. Patient data and clinical symptoms from electronic medical records and Minimum Data Set, COVID-19 status by PCR testing of nasopharyngeal swabs; staff residence from zip codes.	**Higher mortality rates among the frail patients:** Of 389 long-stay residents 146 (37.5%) tested positive for COVID-19 and of these 66 of the 146 (45.5%) were asymptomatic. **Wide variation between nursing units in COVID-19 rates:** Nursing units varied widely (0–90.5%) in percentage COVID-19 positive. Of the COVID-19 positive residents 44 (30.1%) died (22.2% of the moderately frail and 50.0% of the frail; *p* < 0.001). In LCTF units 6% (95% CI 1.04, 1.08) increase in positive COVID-19 tests for each 10% increase in percentage of staff living in communities with high COVID-19 prevalence.
McMichael 2020 [22]	**Progression of COVID-19 epidemic in LTCF:** King County, Washington State, USA. Confirmed COVID-19 case identified 28 February 28 2020.	By 18 March 18 167 confirmed COVID-19 cases (101 residents, 50 HCWs, 16 visitors) epidemiologically linked to the facility. **Hospitalisation rates** **for** **COVID-19 positive residents:** 54.5%, visitors 50.0%, staff 6.0%; 34 deaths in the 101 residents and one in a visitor.
Kennelly 2021 [23]	**National point-prevalence COVID-19 testing programme in Ireland:** residents and staff conducted 18 April to 5 May 2020 in all nursing homes and then if a new COVID-19 case was discovered every two weeks. For 45 nursing homes in Dublin and eastern Ireland complete surveys received from 28 homes (62.2%) for 2043 residents.	**Progression of epidemic:** First laboratory-confirmed community COVID-19 case in Ireland 29 February 2020. In the national survey the confirmed COVID-19 rate for residents 40.8% (25% asymptomatic); case fatality rate 25.8%; for staff confirmed rate 33.6% within the first 28 days of an outbreak and 28.9% subsequently (27.6% asymptomatic).
Garibaldi 2021 [25]	**Retrospective cohort study in 5 hospitals:** 832 consecutive COVID-19 admissions 4 March to 24 April 2020, five hospitals Maryland and Washington, DC.	**Progression to more severe COVID-19:** 787 admitted with mild to moderate disease, 45 with severe disease (WHO scale) [23]. At discharge 523 (63%) had experienced mild to moderate disease, 171 (20%) severe disease and 131 (16%) died. Progression to severe disease or death rapid and occurred in 181 (60%) by day 2 and 238 (79%) by day 4. Progression to severe disease or death correlated with BMI, respiratory symptoms, respiratory rate, C-reactive protein (CRP) level, albumin level, and temperature > 38.0 °C and for those 60 to 74 years a detectable troponin level. Older age and nursing home residence were associated with high comorbidity levels and risk of death.
Telle 2021 [26]	**National study in Norway:** all 8569 individuals who tested positive for SARS-CoV-2 by end of June 2020.	**Outcomes for ≥90 year olds compared to <50 year olds:** risks of hospitalisation (RR = 9.5; 7.1, 12.7) and death (RR = 607.9; 145.5, 2540.1) much higher and risk of death for nursing home residents was higher (RR = 4.2; 3.1, 5.7)
Tarteret 2020 [27]	**Comparison of 3 nursing homes in France:** 2 hospital-dependent nursing homes in France with permanent physicians and connections with infection prevention and control departments and a nursing home without permanent physicians, infection control practitioner home or direct connection with a general hospital.	**Mortality rates:** During first 3 months of the COVID-19 outbreak 224/375 (59.7%) residents classified as COVID-19 and 57/375 (15.2%) died with rates of 6.6% in the hospital-dependent homes and 25.8% in the non-hospital-dependent home, OR = 0.20 (0.11, 0.38; *p* = 0.001). **Risk factors for mortality:** in COVID-19 patients during first 3 weeks of outbreak lower if received a daily clinical examination OR = 0.09 (0.03, 0.35; *p* = 0.01), three vital signs measured daily OR = 0.06 (0.01, 0.30; *p* = 0.001) and prophylactic anticoagulation OR = 0 (0.00, 0.24; *p* = 0.001).
Gopal 2021 [28]	**COVID-19 outbreak sizes in 713 LCTFs:** California to 1 May 2020.	**Outbreak sizes:** 12.7 times larger in for-profit than non-profit LCTFs (p <.001). Higher ratings for approved Centers for Medicare and Medicaid services correlated with fewer infections in residents (*p* < 0.001) and staff (*p* < 0.05).
Castriotta 2020 [29]	**Community COVID-19 mortality rates in Italy:** Friuli Venezia Giulia region, northern Italy.	**COVID-19 mortality rates higher in older seniors:** for those 70–79 SMR = 16.13 (95% CI 9.73, 26.74) and for those ≥80 SMR = 35.58 (95% CI 21.77, 58.15) compared to those <70 years. No significant differential mortality for seniors in nursing homes. **Mortality variations between provinces:** Standardised mortality rates as of 23 June 2020 varied from high of 2.92 (95% CI 2.88, 2.97) in Lombardia, and 1.95 (95% CI 1.64, 3.30) in Valle D’Aosta to a low of 0.71 (95% CI 0.68, 0.74) in Veneto, and in central Italy SMR = 0.13 (95% CI 0.11, 0.17) in Umbria and 0.26 (95% CI 0.24, 0.28) in Lazio, with unexplained local transmission patterns.

**Table 5 geriatrics-06-00048-t005:** Interventions to increase influenza and pneumococcal vaccination rates in seniors and in HCWs.

Author, Date	Setting	Interventions	Outcomes or Observations
Thomas 2018 [39]	**Systematic review of 61 RCTs to increase influenza vaccination rates** in ≥60 years	(1) Increase demand from individuals, (2) increase vaccine access, (3) increase provision. 38% of RCTs assessed low risk of bias for randomisation, 11% allocation concealment, 44% blinding, 51% missing data, 0% selective reporting; overall evidence low quality.	**Three types of interventions to increase influenza vaccination rates:** (1) 41 RCTs (767,460 participants) increasing patient demand: invitations by clinic receptionists (OR 2.72; 1.55 to 4.76); nurses or pharmacists educated, and nurses vaccinated patients (OR 152.95; 9.39 to 2490.67); medical students counselled patients (OR 1.62; 1.11 to 2.35); multiple recall questionnaires (OR 1.13; 1.03 to 1.24). (2) 8 RCTs increasing vaccine access (9353 participants); invitations during home visits (OR 1.30; 1.05 to 1.61), free vaccine (OR 2.36; 1.98 to 2.82), invitations during consultations with patient groups. (3) 15 RCTs tested interventions with HCWs or medical systems; payments to physicians (OR 2.22; 1.77 to 2.77), reminding physicians to vaccinate all patients (OR 2.47; 1.53 to 3.99); clinic posters of vaccination rates and encouraging doctor competition (OR 2.03; 1.86 to 2.22); chart reviews benchmarking to rates of top 10% of physicians (OR 3.43; 2.37 to 4.97).
Gravenstein 2017 [40]	**823 nursing homes in USA**, Medicare-certified (92,269 residents; 75,917 ≥ 65 years)	**409 homes randomised to high-dose influenza vaccination**, 414 homes to standard-dose vaccine	**Respiratory-related hospital admissions rate:** significantly lower (3.4% over 6 months) in homes whose residents received high-dose influenza vaccines vs. 3.9% in standard-dose influenza vaccines; adjusted (RR) = 0.873 (0.776 to 0.982; *p* = 0.023).
**Interventions to Increase Seniors’ Pneumococcal Vaccination Rates**			
Naito 2020 [41]	**Japan, national vaccination campaign,** 2014	**Public subsidy for pneumococcal vaccination** (PPV23) for ≥65 years	**Vaccination rates:** 0% in 2009, 10% in 2011, 40.6% after campaign in 2015, 74% 2018. Child vaccination programme included PCV7 then PCV13 resulting in increase in community prevalence of serotypes 8, 9N and 12F (which comprise 40% of serotypes causing Invasive Pneumococcal Disease (IPD) in elderly). However, these serotypes are included in PPV23 which thus provides protection to elderly.
Murakami 2019 [42]	**Japanese Health Ministry survey influenza vaccine coverage** all municipalities (*n* = 1741); 1010 municipalities (58.0%) responded	Direct mail offer of subsidised PPV23 vaccination	Median PPV23 coverage for ≥65 years for responding municipalities 2016 41.8%. **Differences in response rates:** 18.7% higher in municipalities which sent direct mail notification to targeted adults. Rate decreased by 3.02% for every ¥1000 increase in out-of-pocket costs to individuals and coverage inversely related to municipality unemployment rates and average per capita income.
**Interventions to Increase Influenza Vaccination Rates of Health Care Workers in LCTFs**			
Thomas 2013 [46], 2016 [47]	**Systematic review of four c-RCTs** and one cohort study (*n* = 12,742) of influenza vaccination for HCWs caring for individuals ≥ 60 years of age in LTCFs. Studies similar in study populations, interventions and outcome measures [48,49,50,51].	**Vaccination offered to residents and HCWs in intervention arms** and usual care in control arms. Bias in studies due to attrition, lack of blinding, contamination in control groups and low rates of vaccination coverage in intervention arms. GRADE quality assessments downgraded for all outcomes due to serious risk of bias.	**Laboratory-proven influenza:** HCW influenza vaccination in LTCFs may have little or no effect on number of residents who develop compared with those living in care homes where no vaccination offered (RD 0 (95% CI −0.03 to 0.03)) (2 studies, 752 participants; low quality evidence); **Lower respiratory tract infection:** HCW vaccination probably leads to reduction in residents from 6% to 4% (RD −0.02 (95% CI −0.04 to 0.01)) (one study, *n* = 3400 people, moderate quality evidence); **Number of residents admitted to hospital for respiratory illness:** HCW vaccination programmes may have little or no effect on (RD 0 (95% CI −0.02 to 0.02)) (one study *n* = 1059; low quality evidence). **Deaths from lower respiratory tract infection:** Data not combined (two studies, *n* = 4459) or all cause deaths (four studies, *n* = 8468). Very low quality of evidence because direction and size of difference in risk varied between studies and uncertainty about the effect of vaccination on these outcomes.

**Table 6 geriatrics-06-00048-t006:** CDC recommendations for nursing home residents with acute respiratory illness symptoms when SARS-CoV-2 and influenza viruses are circulating.

⮚ Ask all residents daily if they have respiratory illness symptoms, daily temperatures, any signs or symptoms. ⮚ Test for SARS-CoV-2 by nucleic acid detection OR by SARS-CoV-2 antigen detection assay (lower sensitivity) so confirm antigen test with SARS-CoV-2 nucleic acid detection assay. ⮚ If a new SARS-CoV-2 infection is identified in a nursing home promptly test all residents. ⮚ Test for influenza by rapid influenza nucleic acid detection assay OR rapid influenza antigen detection assay (lower sensitivity) so confirm antigen test with influenza nucleic acid detection assay. ⮚ For symptomatic residents use all recommended PPE with suspected SARS-CoV-2 infection, move to a single room, no new roommates, move to the COVID-19 care unit when confirmed by SARS-CoV-2 testing. ⮚ Promptly notify health department for further investigation of suspected or confirmed case of SARS-CoV-2 or influenza in a resident or a healthcare person, a resident with severe respiratory infection resulting in hospitalization or death; or ≥ 3 residents or HCP with new-onset respiratory symptoms within 72 hours of each other. ⮚ Move all residents with confirmed SARS-CoV-2 infection to a dedicated COVID-19 care unit. ⮚ Residents found to have SARS-CoV-2 and influenza virus co-infection should be placed in a single room on the dedicated COVID-19 unit or housed with other co-infected residents on that unit. These residents should continue to be cared for using all recommended PPE for the care of a resident with SARS-CoV-2 infection. ⮚ Place residents with confirmed influenza in a single room, or with other residents with only influenza, and if unable to move resident, use measures to reduce transmission to roommates (e.g., physical barriers, antiviral chemoprophylaxis) and droplet precautions.

**Table 7 geriatrics-06-00048-t007:** Preventive COVID-19 interventions and outcomes in LTCFs.

Author, Date	Setting	Interventions	Outcomes or Observations
Goto 2021 [67]	**US Veterans’ Affairs** Midwest HealthCare Network	(1) Admit patients from hospitals or communities with no COVID-19 cases. (2) Quarantine admissions in single-patient rooms 14 days. (3) Daily screening for temperature, symptoms. (4) Only visitors critical to care-giving. (5) No temporary staff. (6) Hand and respiratory education. (7) Supervised by full-time infection on-site preventionists and infectious disease specialists.	**Minimal COVID-19 infections:** All residents from 6 March to 1 September 2020 reverse-transcriptase polymerase chain reaction (RT-PCR) for SARS-CoV-2 negative; 4 employees positive and asymptomatic and furloughed.
Vijh 2021 [68]	**75 LTCFs in British Columbia**	(1) Symptom assessment, testing all residents and staff; contact tracing; isolation of high risks. (2) Universal personal protective equipment (PPE) all staff; contact and droplet precautions all COVID-19 cases (confirmed, suspected or exposed) and residents with significant exposure. (3) COVID-19 mobile team provided assessment and education. (4) No admissions or community discharges. (5) Residents restricted to rooms; staff cohorted to wards; COVID-19 residents cohorted to rooms. (6) Enhanced cleaning rooms, common spaces, high-touch surfaces. (7) Check-in with staff provision additional staff/resources. Daily	**Initial outbreak:** 28 February to 30 May 2020, 18 (24%) LTCFs had at least 1 documented exposure from a COVID-19 case (total *n* = 165 staff and 110 residents). During the two weeks after outbreak significant increase in COVID-19 incidence RR = 1.07 (1.03 to 1.11; *p* < 0.001). **Results 14 days and onwards after interventions were implemented:** significant decrease in cases RR = 0.68 (0.62 to 0.75; *p* < 0.001) and 27% decrease in incidence rate every 2 days RR = 0.73 (0.67 to 0.80; *p* < 0.001).
Telford 2020 [69]	24 LTCFs, Fulton County, Georgia, which had 85% of COVID-19 positive residents of all LTCFs in the county	**CDC COVID-19 tool indicators assessed prevalence** [59,60]: **11 LTCFs with higher prevalence:** (1310 residents, 817 cases) average infection rate 62% (range 46–74%), 196 hospitalisations, 124 deaths. **13 LCTFs with lower prevalence:** (1270 residents, 187 cases) average infection rate 15% (range 1–33%), 51 hospitalisations, 38 deaths.	**Prevention implementation in lower COVID-19 prevalence LTCFs compared to higher prevalence LTCFs:** 69% implementation of hand hygiene indicators (55% in higher prevalence group), 77% disinfection indicators (36%), 74% social distancing indicators (54%), personal protective equipment indicators 72% (41%), 82% screening indicators (64%).
Belmin 2020 [70]	**17 nursing homes in France compared to national survey of 9513 LTCFs** (385,290 staff; 695,060 residents)	**17 nursing homes in which 794 staff members voluntarily confined themselves to the facility** with their 1250 residents.	**17 nursing homes in which staff confined themselves with patients:** 1/17 (5.8%) homes had 5 (0.4%) COVID-19 resident cases; 5 (0.4%) deaths; confirmed or possible COVID-19 in 12 (1.6%) staff members. **National survey of nursing homes:** 30,569 (4.4%) COVID-19 resident cases (*p* < 0.001); self-confinement of 31,799 (4.6%) residents; 12,516 (1.8%) resident deaths (OR = 0.22 (95% CI 0.09 to 0.53; *p* < 0.001); confirmed or possible COVID-19 in 29,463 staff members (7.6%) (*p* < 0.001).
Brown 2021 [71]	**618 nursing homes Ontario, Canada** (78,607 residents) = 99% of all 623 homes	**Assessment of effect of crowding** (crowding index assessed single-bedded to four-bedded rooms).	29 March to 20 May 2020 5218 (6.6%) residents developed COVID-19 infection; 4496 (86%) of infections occurred in only 63 (10%) homes; 1452 (1.8%) residents died of COVID-19 infection to May 20 2020. **Low crowding index homes:** COVID-19 incidence 4.5%; mortality 1.3%. **High crowding index homes:** COVID-19 incidence 9.7% (*p* < 0.001), mortality 2.7%; (*p* < 0.001).
Jones 2021 [73]	**623 LCTFs in Ontario, Canada**	**Policy to prevent staff working in multiple LTCFs**	**Before policy:** 266 (42.7%) homes had staff working in at least 1 other home. **After policy instituted**: 79 (12.7%) homes had staff working in 1 other home (decrease 70.3% (*p* < 0.001)); average number of connections between homes declined 3.90 to 0.77 (decrease of 80.3%, *p* < 0.001)

**Table 8 geriatrics-06-00048-t008:** Interventions to decrease respiratory disease transmission using masks, hand washing, isolation rooms, decreasing surface contamination and identifying communities at risk.

Interventions to Decrease Respiratory Disease Transmission Using Masks, Hand Washing, and Isolation			
Author, Date	Setting	Interventions	Outcomes or Observations
Jefferson 2020 [75]	Hospital wards in high-income countries, suburban schools, and inner cities in low-income countries.	**Comparison of medical or surgical masks to no masks:** 8 c-RCTs, 1 RCT (2 trials with healthcare workers and 7 in the community).	**Low certainty evidence mask wearing may make little or no difference in influenza-like illness (ILI):** compared to not wearing masks RR = 0.99 (0.82, 1.18; 6 trials, 3507 participants). **Moderate certainty evidence mask wearing probably makes little or no difference to laboratory-confirmed influenza**: compared to not wearing masks RR = 0.91 (0.66, 1.26; 6 trials; 3005 participants).
Jefferson 2020 [75]	Comparison of respirators and masks.	**Comparison of N95/P2 respirators to medical/surgical masks.**	**Clinical respiratory illness:** very low certainty evidence: RR = 0.70 (0.45, 1.10; 3 trials; 7779 participants); **ILI:** low-certainty evidence: due to imprecision and heterogeneity RR = 0.82 (0.66, 1.03; 5 trials; 8407 participants); **Laboratory-confirmed influenza:** little or no difference and moderate-certainty evidence: RR = 1.10 (0.90, 1.34; 5 trials; 8407 participants) with no differences for health care workers (HCWs).
Jefferson 2020 [75]	Hand hygiene studies in schools, childcare centres, homes, and offices.	**Hand hygiene interventions compared to no intervention.**	**Acute respiratory infections (ARIs):** hand hygiene interventions compared to no intervention: 16% relative reduction in number of people with RR = 0.84 (0.82, 0.86; 7 trials; 44,129 participants; probable benefit with moderate-certainty evidence); **ILI:** RR = 0.98 (0.85, 1.13; 10 trials; 32,641 participants; little or no difference with low-certainty evidence); **Laboratory-confirmed influenza:** RR = 0.91 (0.63, 1.30; 8 trials; 8332 participants; little or no difference with low-certainty evidence).
Cheng 2018 [79]	10 residential care homes for the elderly, Hong Kong.	**5 homes randomised to directly observe hand hygiene (DOHH) of residents’ hands by hand hygiene ambassador nurses,** and 5 homes randomised to usual care control group. **Intervention:** (515/774 residents participated) hand cleaning by hand hygiene ambassador nurse at two-hourly intervals and also before meals and medication rounds 9 am to 6 pm weekdays; during 8-week intervention samples collected twice weekly immediately before environmental cleaning from communal areas (blood pressure cuff, meal table-top, activity table-top, chair armrest, corridor hand rail, remote TV control), and in staff areas (station table top, computer keyboard and mouse, trolley-top and handle, telephone handle).	**Baseline colonisation:** 33% of 100 samples culture-positive for methicillin-resistant *Staphylococcus aureus* (MRSA); 26% of 100 specimens for carbapenem-resistant *Acinetobacter* species (CRA). **Serial monitoring of colonisation during 2 month intervention:** **MRSA:** present in 79/600 (13.2%) of samples in intervention homes and in 197/600 (32.8%; *p* < 0.001) in control homes, and **CRA**: present in 56/600 (9.3%) of samples in intervention homes and 94/600 (15.7%; *p* = 0.001) in control homes. **Volume of Alcohol Based Hand Rub (ABHR):** consumed/resident/week 3 times higher in intervention group (59.3 ± 12.9 mL) compared with baseline (19.7 ± 12.6 mL; *p* < 0.001) and significantly higher than in control group (23.3 ± 17.2 mL; *p* = 0.006). **Hand hygiene compliance:** improved from 27% to 60% during study period, but adherence to WHO Five Moments hand hygiene campaign not sustained, and bacterial contamination occurred with return from hospital care.
Chu 2020 [80]	Systematic review of physical distancing, face masks and eye protection on spread of SARS-CoV-2.	Search of 21 WHO-specific and COVID-19 sources: 172 observational studies; 44 non-randomised studies selected for meta-analysis, no RCTs identified.	**Virus transmission: Lower rates with physical distancing ≥ 1 metre:** (*n* = 10,736, OR = 0.18 (0.09 to 0.38); risk difference (RD) −10.2% (−11.5 to −7.5; moderate certainty); Protection increased with distance RR = 2.02 per metre (*p* = 0.041); moderate certainty). **Lower rates with face mask use:** (*n* = 2647; OR = 0.15 (0.07 to 0.34, RD −14.3%, −15.9 to −10.7; low certainty)). **Lower rates with N95 or similar respirators compared with disposable surgical or cotton masks:** (*p* = 0·090); low certainty. **Lower rates with eye protection:** (*n* = 3713; OR = 0.22 (0.12 to 0.39, RD −10.6%, 95% CI −12.5 to −7.7; low certainty)).
**Interventions and Models of Interventions to Isolate Infected Patients**			
**Author, Date**	**Setting**	**Interventions**	**Outcomes or Observations**
Kim 2020 [81]	**Emergency department,** Chungbuk National University Hospital, Cheongju, South Korea.	27 February to 31 March 2020, 2455 patients assessed for potential COVID-19 and if fever or respiratory symptoms they were screened in triage room, and if indicated COVID-19, test and chest X-ray obtained. Transported on isolation stretcher to CT unit.	Before isolation strategies implemented: emergency department shut down for 2 hours of cleaning 1.6 times/day; after isolation strategies 0.6 times/day.
Cho 2019 [82]	**Model of infection control room for airborne infectious.** Manikin used to assess flow of potentially contaminated air from patient in bed directed to HCW providing care.	Fresh external air flowed over patient’s bed from ceiling vent and vented externally: venting either ceiling vent, single vent under bed, or two vents 1.2m above floor behind patient’s bed. Air flows visualised by fog generator from manikin’s mouth using SF_6_ (sodium hexafluoride).	For HCW 1.4m from patient concentration of SF_6_ with ceiling exhaust 33.1 to 72.7 ppm, with exhaust under bed 25.1 to 34.4 ppm, with dual exhausts in wall behind the bed 21.2 to 24.4 ppm and for two exhausts in the wall either side of bed with a Fan Filter Unit (FFU) with a 0.3-micron pore size HEPA filter (rated 99.997% efficient at retaining particles) concentration 1.4m and 0.9m above the floor was 2.0 to 8.9 ppm, 85.2% lower than without the FFU and for the whole room 79.6% lower than without the FFU.
Kalliomäki 2016 [83]	**Model of model isolation room.**	Air flowed into the room and was exhausted by vents at the top of the room. Air flows were filmed and change in air flows visualised with smoke generator as door opened, manikin entered and doors closed.	During 24 second period as doors opened and manikin entered room the plume of smoke was dragged by the manikin into the room, the plume passed in front of manikin and mixed with the room air and doors closed. More air influx occurred with hinged doors than sliding doors.
Shao 2020 [84]	**Model clean room** with air pressure 15 Pa > anteroom and anteroom 10 Pa > surrounding room.	Manikin walking speed 0 m/s, 0.5 m/s, 1.0 m/s (=fast walking; airflow rates in cleanroom 200 L/s, 400 L/s, 580 L/s; TSI Atomizer 9302 particle generator generated particles 0.5 to 3.0 μm with 25 psi pressure.	Particle concentration in clean room before door opening range 18,519 to 100,482 /m^3^ (meets ISO 14644 specification for classes 6 and 7). With door closed no particles entered cleanroom due to overpressure of 15 Pa; few particles entered with door opening and closing. When manikin entered walking at 0.5 m/s and airflow rate 210 L/s 840,994 particles/m^3^ entered cleanroom; at 400L/s 83,774 particles/m^3^ and at 580 L/s 42,407 particles/m^3^ (meets ISO 14644 Class 7 specification); Particle counts in cleanroom tripled with manikin entering at fast walking speed of 1 m/s: 1,745,142/m^3^ at 210 L/s; 247,580/m^3^ at 440 L/s and 120,417/m^3^ at 580 L/s.
Mousavi 2020 [85]	**Model conversion of patient room to an isolation room** with a temporary anteroom and air purifier.	Two High Efficiency Particulate Air (HEPA) machines drew air from patient room at 1500 m^3^ h^−1^ to exterior yielding 20 air changes/hour and a negative pressure of 2.5 Pa. Particle concentration in patient room < 1000 for particle size 0.3 μm Nebuliser released aerosols at 10^5^ particles/L from manikin’s head lying in bed (1000 above room concentration to simulate a SARS-CoV-2 virus count from a cough).	Marked increase in concentration within patient room with aerosol simulating coughing. Highest migration rate from patient room for particles < 3 μm compared to > 3 μm. Plastic barrier in anteroom without HEPA filters prevented spread of 80% of particles. HEPA filters markedly reduced particle counts in anteroom and hallway.

**Table 9 geriatrics-06-00048-t009:** UVC lights to decrease surface and air contamination in nursing homes.

Author, Date	Setting	Interventions	Outcomes or Observations
Anderson 2017 [98], 2018 [99]	**C-RCT, 9 hospitals, SE USA,** single rooms from which patients had been discharged who had had positive cultures of four target organisms in previous 12 months.	21,395 patients randomised to 4 study arms: (1) quaternary ammonium disinfectant (QUAD) (except bleach for *C. difficile*); (2) UV and QUAD (or UV and bleach for *C. difficile*); (3) bleach; (4) bleach and UV. Randomisation by random-number generator, 99% power to detect 20% decrease in incidence rates, microbiological identification used standard protocols, environmental services personnel trained on use of disinfectants, cleaning protocols, UV lights; compliance with protocols, hand hygiene and cleaning similar across study groups, cleaning compliance 90%; QUAD applied using microfibre cloths (which remove more bacteria than cotton or synthetic fibres).	New patients admitted to rooms which had been occupied by patients who had had methicillin-resistant *Staphylococcus aureus*, vancomycin-resistant *enterococcus*, *C. difficile*, or multidrug-resistant *Acinetobacter*. Patients in the QUAD arm: had 51.3 cases of the targeted organisms/10,000 patient days. Patients in UV + QUAD arm: 33.9 cases/10,000 exposure days, RR = 0.70 (0.50–0.98; *p* = 0.036). Patients in bleach arm: 41.6 cases/10,000 exposure days, RR = 0.85 (0.69–1.04; *p* = 0.116). Patients in UV + bleach arm: 45·6 cases/10,000 exposure days, RR = 0.91 (0.76–1.09; *p* = 0.303). Incidence of *C. difficile* infections: among exposed patients unchanged for UV plus bleach compared to bleach with 30.4 cases vs. 31.6 cases/10,000 exposure days RR = 1.0 (0.57 to 1.75; *p* = 0·997).
Ethington 2018 [97]	**Before-after study of special care unit of a long-term acute care hospital.**	Airborne bacterial colony forming units (CFU)/m^3^ of air were measured in 16 patient rooms, hallway and biohazard room. Ultra-violet germicidal irradiation equipment installed in these locations.	On resampling 81 days later 42% decline in number of airborne bacteria CFU/m^3^ (average 175 vs. 102 CFU/m^3^), rate of infections/month in the home declined from 20.3 to 8.6 (*p* = 0.001), annual number of *Clostridium difficile* cases declined from 8 to 1 (*p* = 0.01), annual number of cases of catheter-associated urinary infections declined from 20 to 9 (*p* = 0.012). No significant decreases in cases of methicillin-resistant *Staphylococcus aureus* (13 vs. 6) or central line-associated bloodstream infections (16 vs. 9).
Buchan 2020 [101]	**Model of 3 meter^3^ room** with air entry top left, air exit top right standard ventilation compared to ultra-violet.	**Far-UVC light** from excimer lamps or light-emitting diodes is safe to use with humans because it generates narrow bandwidth short wavelength UVC (207–222 nm) which does not affect cornea.	**Far-UVC:** Disinfection rates increased by 50–85%. **Far-UVC and high ventilation (8 air changes/hour):** time to achieve 90% reduction in viral count = 6 minutes; 8 air changes/hour results in 99% reduction in 11.5 minutes.

## Data Availability

Not applicable.

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
