# Peer review of "Reducing Morbidity and Mortality Rates from COVID-19, Influenza and Pneumococcal Illness in Nursing Homes and Long-Term Care Facilities by Vaccination and Comprehensive Infection Control Interventions"

_geriatrics, 2021, doi:10.3390/geriatrics6020048_

Round 1
Reviewer 1 Report
Submitted article is original and very interesting. It requires however, significant adjustments, indicated as follows:
- The abstract must be reduced. The number of words must be below 200.
- Introduction - please provide more background to the study.
- Methods - please explain why employed methods are correct for this paper. Also the approach to achieve articles objective should be provided there.
- Discussion and conclusions - I would rather see separate conclusion section from discussion. The findings are written in a bullet form fashion rather than in full sentences.
Author Response
Responses to reviewer 1
|
Reviewers’ recommendations |
Responses |
|
|
Many thanks for your helpful suggestions. I have implemented all of them.
|
|
|
The second reviewer requested that much of the material be presented in tabular form and, therefore, the line and reference numbering have changed. |
|
1. The abstract must be reduced. The number of words must be below 200.
|
Abstract has been rewritten and is now < 200 words |
|
2. Introduction - please provide more background to the study.
|
I have described the key pathogens involved in respiratory illnesses and linked them to the purposes of the article |
|
3. Methods - please explain why employed methods are correct for this paper. Also the approach to achieve articles objective should be provided there.
|
I conducted a new literature search on 9 March 2021 and have explained the method of article selection for this paper |
|
4. Discussion and conclusions - I would rather see separate conclusion section from discussion. The findings are written in a bullet form fashion rather than in full sentences.
|
The Discussion and Conclusion sections are now separate and have been rewritten. |
Reviewer 2 Report
Dear Author,
Thank you very much for your efforts on this topic. The preparation and presentation of evidence-based data on the impact of vaccination and infection control measures on nursing home residents is very important.
For better readability and thus the accessibility of your readership, as well as for sufficient consideration of uncertainties in the interpretation and final transformation of study results to everyday clinical practice, here are a few comments.
Abstract:
The abstract is too long and should be shortened. Results such as the report on the French study should rather be integrated into the results section to this extent. It should be avoided to give the impression of a consensus statement with only one author.
It is important to put the selected evidence into the current context, so the recommendation is made to send all new admissions into a 14-day quarantine. Here, for example, newer test concepts and also shorter quarantine times should be examined. As in this case, the other recommendations should also be checked to ensure that they are up to date.
Introduction:
Table 1 is not in correct format.
The numbering should not 1. and 1.1. when you do not come to point 1.2.
Materials and Methods
Table 2. Please sort the serch terms on the right side and not in the middle. Please provide subheadings within the table for your main search questions.
Table 2 search term: Sars-CoV-2 is missing. It is not mentioned in the methods if search terms where used for whole text search or for title and abstracts.
Results:
Please provide your text-information from section 3.1-3.5 in one table structured according to your subheadings and shorten the text as much as possible.
Please provide your text-information from section 4.1-4.4 in one table structured according to your subheadings and shorten the text as much as possible.
Please provide your text-information from section 4.5 in one table structured according to your subheadings and shorten the text as much as possible. Please provide in all studies you mention the date of publication.
Please provide your text-information from section 4.6-4.9 in one table structured according to your subheadings and shorten the text as much as possible. Please provide in all studies you mention the date of publication.
Discussion
Please avoid results in the discussion. In the most sections of the discussion are results. Please provide new subsections (in the results) for the additional data you show in the discussion.
In order to counteract the impression of presenting a past-related, partly outdated evidence, please relate the results of your search in your discussion, especially the recommendations mentioned in the abstract (in the form "there is strong evidence"), to 1. the most current guidelines / recommendations of professional societies / responsible government agencies 2. new test concepts 3. new foreseeable developments.
Author Response
Responses to reviewer 2
|
Suggestions of Reviewer 2 |
Responses and changes |
|
Thank you very much for your efforts on this topic. The preparation and presentation of evidence-based data on the impact of vaccination and infection control measures on nursing home residents is very important. |
Thank you for your very helpful suggestions. I have implemented all of your suggestions. There is a lot of research and your suggestion about presenting it in tables is very helpful. I reread the studies in detail as I composed the tables. The article text has been substantially rewritten and summarises the key findings in each table. |
|
For better readability and thus the accessibility of your readership, as well as for sufficient consideration of uncertainties in the interpretation and final transformation of study results to everyday clinical practice, here are a few comments. |
|
|
Abstract: The abstract is too long and should be shortened. Results such as the report on the French study should rather be integrated into the results section to this extent. It should be avoided to give the impression of a consensus statement with only one author. |
The abstract has been rewritten and is now < 200 words
|
|
It is important to put the selected evidence into the current context, so the recommendation is made to send all new admissions into a 14-day quarantine. Here, for example, newer test concepts and also shorter quarantine times should be examined. As in this case, the other recommendations should also be checked to ensure that they are up to date |
I have verified the recommendations in the articles are in line with the latest WHO and CDC documents. Of the references 18 are from 2021, 24 from 2020, and 7 each from 2019 and 2018 but many key articles are from earlier dates |
|
Introduction: Table 1 is not in correct format. The numbering should not 1. and 1.1. when you do not come to point 1.2. |
Corrected |
|
Materials and Methods: Table 2. Please sort the serch terms on the right side and not in the middle. Please provide subheadings within the table for your main search questions. Table 2 search term: Sars-CoV-2 is missing. It is not mentioned in the methods if search terms where used for whole text search or for title and abstracts. |
I performed a new search on 9 March 2021 and included Sars-CoV-2 as a search term. The table has been rewritten. |
|
Results: Please provide your text-information from section 3.1-3.5 in one table structured according to your subheadings and shorten the text as much as possible. |
A table has been constructed and the key findings summarised briefly in the text. |
|
Please provide your text-information from section 4.1-4.4 in one table structured according to your subheadings and shorten the text as much as possible. |
A table has been constructed and the key findings summarised briefly in the text. Authors’ dates are provided in the table |
|
Please provide your text-information from section 4.5 in one table structured according to your subheadings and shorten the text as much as possible. Please provide in all studies you mention the date of publication. |
A table has been constructed and the key findings summarised briefly in the text. Authors’ dates are provided in the table |
|
Please provide your text-information from section 4.6-4.9 in one table structured according to your subheadings and shorten the text as much as possible. Please provide in all studies you mention the date of publication. |
A table has been constructed and the key findings summarised briefly in the text. Authors dates are provided in the table |
|
Discussion Please avoid results in the discussion. In the most sections of the discussion are results. Please provide new subsections (in the results) for the additional data you show in the discussion. |
The Discussion and Conclusions sections are now separate and do not include any results. |
|
In order to counteract the impression of presenting a past-related, partly outdated evidence, please relate the results of your search in your discussion, especially the recommendations mentioned in the abstract (in the form "there is strong evidence"), to 1. the most current guidelines / recommendations of professional societies / responsible government agencies 2. new test concepts 3. new foreseeable developments. |
I have summarised the evidence and provided the key numerical results for the studies. |
Round 2
Reviewer 1 Report
The article is on interesting subject and adds to current pandemic literature.
I wish the author(s) good luck with further research.
Author Response
Response to Reviewer 1’s second round of comments
Many thanks for your comments:
“The article is on interesting subject and adds to current pandemic literature.”
“I wish the author(s) good luck with further research.”
I have read each sentence in the text and the tables and made multiple changes to improve clarity.
The first reviewer suggested deleting one table and simplifying the other tables and that has been done.
Reviewer 2 Report
I have read through the revised version of the manuscript. The manuscript has improved somewhat through editing in the introduction, abstract, discussion and conclusion.
But the recommended revisions to the table seem to have resulted in a predominance of copy paste behaviour. The attempt to "squeeze" all currently available information relatively unfiltered into the review tables seems to have failed.
Major revision is needed in the following parts:
1.) Every single table should be reworked and shortend, if nessecary with additional columns.
2.) It does not seem necessary to publish the contents of, for example, the CDC in tabular form (again).
Author Response
Response to Reviewer 2’s second round of comments
Many thanks for your comments:
“I have read through the revised version of the manuscript. The manuscript has improved somewhat through editing in the introduction, abstract, discussion and conclusion.
But the recommended revisions to the table seem to have resulted in a predominance of copy paste behaviour. The attempt to "squeeze" all currently available information relatively unfiltered into the review tables seems to have failed.
Major revision is needed in the following parts:
1.) Every single table should be reworked and shortend, if nessecary with additional columns.
2.) It does not seem necessary to publish the contents of, for example, the CDC in tabular form (again).”
Response
I have read every sentence in the text and the tables and made multiple changes to improve clarity.
For the tables I agree with your comment about copy pasting. I have revised every entry in every table and shortened and clarified each sentence as much as possible. I went back to the original articles to clarify details. Then I went over each table again to look for possible simplifications.
I have deleted the CDC table.
Round 3
Reviewer 2 Report
Dear Author,
Thank you very much for your efforts on this topic. The preparation and presentation of evidence-based data on the impact of vaccination and infection control measures on nursing home residents is very important.
The manuscript is much more readable and has improved much.
I have only a few comments, please see below. I hope these will be inculded soon, so that the article coul be puublished in the next weeks.
Introduction:
Please proof in line 56 if „genome“ is correct.
Results
Line 306 Page 14: Please provide the results of „Interventions to increase influenza vaccination rates in seniors“ in and „Interventions to increase pneumococcal vaccination rates in seniors“ the same table layout like „Intervention to increase influenza vaccination rates of health care workers in LCTFs“
Discussion
1.) Text: „A large RCT with health workers to assess the effects of optimal vaccination, PPE use, hand washing 479 and social distancing on the patients they care for is needed.“
Comment: From a clinical point of view, what should a randomised trial look like? Should FFP2/PPE2 masks be compared with surgical masks? This is not possible in many countries for ethical reasons. Therefore, this proposal should be reconsidered or formulated more cautiously.
2.) Text: Automatic camera observation of hand-486 washing, gloving, mask wearing, eye protection and gown wearing needs to be implemented because current studies 487 demonstrate these interventions are often inadequately used. 488
Comment: In most countries, there will be ethical concerns about camera monitoring of hand washing. In addition, the staff do not work as prison inmates and the individual presumption of innocence applies first.
Rather, regular testing of staff and regular training to increase compliance and high vaccination coverage should be increased among staff in vulnerable facilities (e.g. LTC, nursing homes), as well as among those living there. For example, increased vaccination coverage results in lower transmission rates (for COVID-19).
Please review your recommendation.
Conclusion
1.) Text: Line 508: „….new admissions quarantined in single-patient rooms for 14 days and monitored“
Comment: This recommendation should in any case take into account current developments, e.g. double vaccination of new nursing home residents. Do they really have to go to quarantine every 14 days? Should this be recommended as a conclusion?
It would be better to refer to a sufficiently long quarantine according to the current official regulations. After all, these can change constantly, e.g. depending on the vaccination situation and the emergence of new variants.
2.) Text: Line 509: „Only visitors critical to care-giving admitted, restricting 509 residents to their rooms including for meals, and cohorting staff to provide care only on specific units. The plans need 510 to be implemented by dedicated teams of infection control specialists for the entire group of nursing homes in a com-511 munity and the rapidity and completeness of implementation closely monitored.“
Comment: The same here:
Please formulate it in a slightly softer way and mention the possibility of new developments (vaccinations, changing requirements on the part of government agencies) (see previous comment).
In some countries, for example, daily contact with nursing home residents is already allowed again, provided that all residents are vaccinated twice. Table 2 search term: Sars-CoV-2 is missing
3.) Text: Line 514„Higher crowding indices (4 patients/room compared to single rooms) correlate with higher infection rates and 514 existing homes need to be restructured and new homes to be built to provide a room for each patient with their own 515 bathroom and toilet and an anteroom at the entrance to their room for washing and personal protective equipment so 516 that staff and visitors can prevent transmission of infection.“
Comment: Here, too, please use a somewhat weaker formulation. For example:
"The structural conditions should be adapted in order to be able to react more quickly to new outbreaks and new pandemic situations, e.g. due to new variants of Sars-CoV-2 or other rapidly transmissible pathogens.“"A room for each patient with their own bathroom and toilet and an anteroom at the entrance to their room for washing and personal protective equipment so that staff and visitors can prevent transmission of infection.
4.) Text: Line 536: „Automatic monitoring by cameras of standard infection reduction strategies 537 with personal protective equipment (handwashing, mask wearing, gloving and eye protection) and especially of the 538 WHO five steps in the prevention of transmission of infections would be most useful.“
Comment: Please do not use autom
Please do not refer to camera surveillance for ethical and scientific reasons. It may be good for training purposes in training situations, but for everyday life, additional staff is needed for the evaluation of the surveillance, etc. See also comment on discussion 2) Text. Furthermore, you do not cite any study that proves the benefit of camera surveillance.
You could write that camera surveillance might bring an additional benefit, but without evidence in the text this is not meaningful either.
Abstract:
Text: „,; restrict visitors, … residents restricted to meals in rooms;“
Comment: In the case of very restrictive measures, it should be added that they must adapt to new scientific findings and government requirements.
Why should dually vaccinated nursing home residents no longer be allowed to eat together in a large common room?
Therefore, I would put this statement into perspective and rather write. "Depending on the vaccination situation and the current risk situation, visiting restrictions and meals in the residents' own rooms may be necessary.
